# MMSeg: Multi-Modal and Multi-View Driven Semantic Enrichment for Training-Free Image Prompt Segmentation

## Abstract

Rapid development of vision foundation models has fueled interest in training-free image segmentation utilizing image prompts. Current methods typically involve a single image and its corresponding mask as references, relying on high-level feature similarity to generate point prompts for subsequent segmentation. However, these approaches suffer from inaccurate target localization and suboptimal mask quality. In response to these limitations, we propose **MMSeg**, a training-free **M**ulti-modal and **M**ulti-view image prompt **Seg**mentation framework. MMSeg enhances semantic information by diversifying references through two key components: visual localization augmented by diffusion prior and multi-view cues, alongside text-driven localization from generated pseudo-labels. By leveraging segmentation consistency across multi-view images and complementary strengths of multi-modal cues, these modules facilitate precise target localization. Furthermore, a consensus-oriented mask proposer is devised to filter and refine mask proposals. Experimental results demonstrate the competitive performance of MMSeg, achieving 95.1% mIoU on the PerSeg dataset, 87.4% on the FSS dataset, and 52.8% on the COCO-$20^i$ dataset.

## 1 Introduction

Image segmentation is a fundamental task in computer vision and critical to perform scene understanding(Brar et al., 2025). It is widely applied in robotics (Jiang et al., 2025), autonomous driving (Shoeb et al., 2025), and medical imaging (Liu et al., 2024a). Recent years have witnessed the emergence of vision foundation models (Kirillov et al., 2023; Ravi et al., 2024; Caron et al., 2021; Oquab et al., 2024; Radford et al., 2021; Zou et al., 2023), which have propelled progress in semantic segmentation by enabling the identification of infinite object categories or visual concepts. In practice, performing segmentation with vision foundation models typically requires appropriate prompts tailored for each image. These prompts can be text descriptions or visual indicators, such as bounding boxes, scribbles, or points. However, providing effective prompts requires either multi-round interaction or professional expertise, which degrades generalization to unknown domains and limits automatic segmentation of batch data.

Recent studies have investigated SAM-based solutions (Kirillov et al., 2023; Ravi et al., 2024) that enable batch segmentation without the need for carefully crafted prompts. These methods generally fall into two categories: those using an image and its corresponding mask as reference, and those employing text as reference, as illustrated in Fig. 1(a) and Fig. 1(b) of the left panel. Among visual-only approaches, PerSAM (Zhang et al., 2024a) is a pioneering method that supports segmentation of customized visual concepts by generating point prompts for SAM through feature matching. Matcher (Liu et al., 2024b) further enhances the quality of generated point prompts and extends SAM's capabilities to more general segmentation tasks. As a text-based approach, ESC-Net (Lee et al., 2025) automatically generates point prompts within the image by leveraging the correlation between text and images, incorporating textual guidance to refine mask generation. However, relying on a single modality for prompting presents inherent limitations in practical scenarios, underscoring the need for more robust approaches that integrate multiple modalities to provide complementary semantic cues and improve segmentation accuracy.

Figure 1: Illustration of our motivations. Left panel: From left to right, we summarize and visualize three paradigms for reference-based image segmentation. Right panel: we provide intuitive analyses on the failure cases of previously adopted paradigms. Meanwhile, we showcase the robust segmentation achieved by the proposed framework driven by image prompts and "hacked" text prompts.

This paper identifies two key issues for improving image-prompt-based segmentation: accurate object localization for point prompts and mask quality optimization. As illustrated in the right panel of Fig. 1, existing methods confront the following issues: 1) Visual-only methods frequently suffer from granularity misalignment and imprecise localization due to reliance on high-level features extracted by models such as DINOv2 (Oquab et al., 2024) or SAM. 2) Text-only methods effectively mitigate granularity inconsistencies but face difficulties caused by linguistic ambiguity or limited descriptive capacity, particularly when targets exhibit subtle or complex attributes that are hard to precisely describe.

In response, we propose a **M**ulti-modal and **M**ulti-view **Seg**mentation framework (**MMSeg**), which integrates pseudo-class generation into the image-prompt segmentation pipeline. To improve localization, we enrich feature diversity using diffusion priors, multi-view image augmentation, and pseudo-label generation. For mask optimization, we first sample point prompts to generate candidate masks, then filter and merge them using a multi-step consensus-oriented pipeline. Experimental evaluations demonstrate that MMSeg significantly reduces both localization and segmentation errors compared to prior methods. Our main contributions are summarized as follows:

- We propose MMSeg, multi-modal and multi-view driven semantic enrichment for training-free image prompt segmentation. By further enhancing visual features and introducing pseudo class generation, our approach enhances semantic expression, enabling more accurate targeting and high-quality mask generation.

- We introduce three key components: Visual Localization augmented by Diffusion prior and Multi-view cues (VLDM) for visual localization, Text-driven Localization from Generated Pseudo-labels (TLGP) for text-driven localization, and Consensus-Oriented Mask Proposer (COMP) for mask optimization. These modules collaboratively alleviate the limited richness of feature representation derived from a single image, mitigate the reliance on high-level features, and improve mask quality of the generated masks.

- Extensive experiments show our method achieves strong performance on multiple datasets and tasks. It supports both one-shot semantic segmentation and generalizes well to one-shot part segmentation. Comprehensive ablation studies validate the contribution and efficacy of each component in our framework.

## 2 METHOD

As illustrated in Fig. 2, the proposed MMSeg framework leverages a single reference image to segment corresponding regions in the test image based on shared semantics. It comprises two main components: multi-modal localization and prompt generation, as well as mask proposal optimization. The localization employs a feature matching pipeline with visual and textual branches to extract feature similarity maps. A cascaded matching process gathers multi-modal point sets that guide precise target localization. Initial mask proposals are generated using the Segment Anything Model (SAM) and refined with a Consensus-Oriented Mask Proposer (COMP) strategy. The optimized masks are then used in SAM for accurate segmentation of regions in the test image that correspond to the reference image mask.

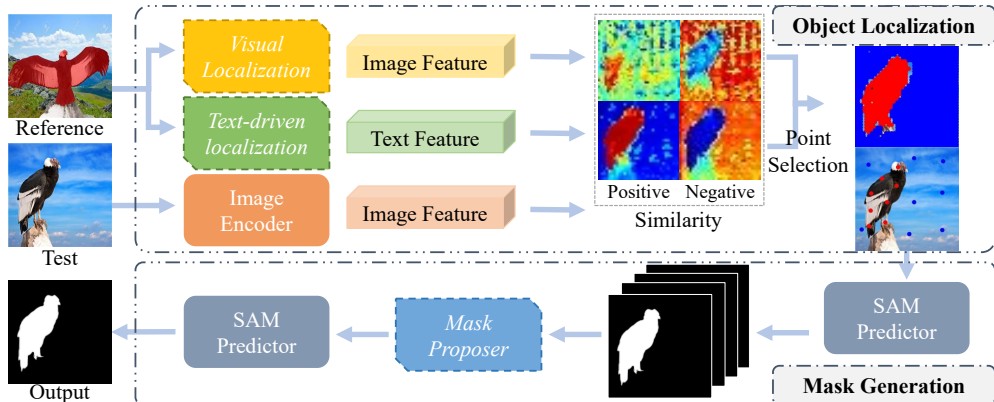

Figure 2: Overall pipeline of the proposed training-free segmetation framework, MMSeg. It consists of two main parts: object localization and mask generation. The localization part in the upper panel contains two independent branches, including VLDM and TLGP. In the rightmost two images, we use red and blue points to represent positive and negative point prompts, respectively. The mask generation part in the lower panel implements a COMP for accurate mask proposal refinement.

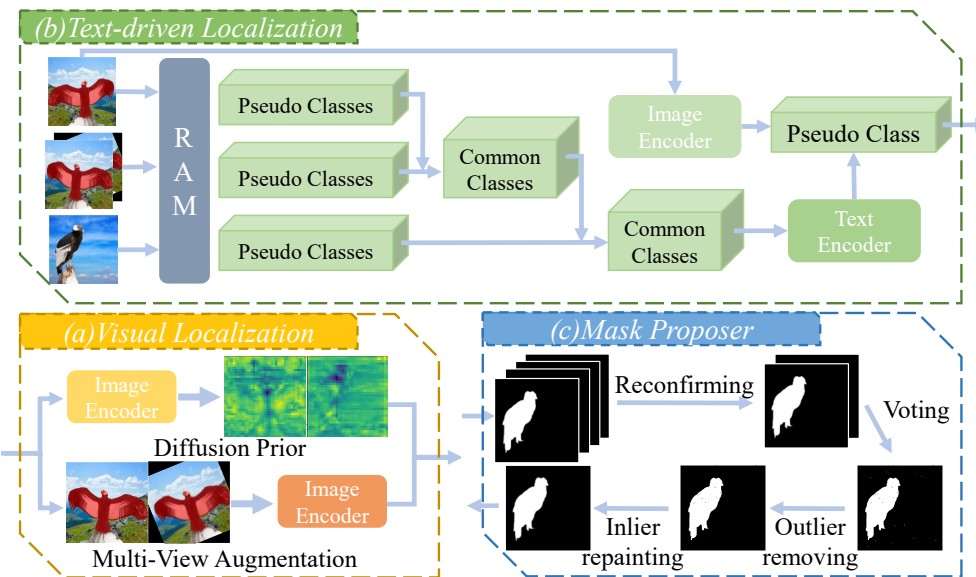

Figure 3: Detailed schematic diagram of VLDM for visual localization, TLGP for text-driven localization, and COMP for mask proposal generation.

## 2.1 VISUAL LOCALIZATION AUGMENTED BY DIFFUSION PRIOR AND MULTI-VIEW CUES

Most methods neglect the crucial role of low-level and diverse features in improving segmentation performance (Tang et al., 2024; Suo et al., 2024). To exploit target location cues embedded in visual prompts, we propose to augment visual features based on leveraging high-level image features. Specifically, we design a visual localization branch enhanced by diffusion priors and multi-view augmentation, as depicted in Fig. 3(a). This branch integrates low-level features extracted through diffusion models and diverse features obtained from multi-view augmented images.

**Diffusion prior**. Inspired by IPSeg, we utilize the prior knowledge embedded in pretrained diffusion models to extract low-level image features, thereby compensating for the tendency of high-level features to overlook fine-grained image details. However, we observe that the stability of diffusion priors is sensitive to the extraction location and the number of diffusion timesteps, rendering the method vulnerable to hyperparameter variations. To overcome the limitation, we introduce CleanDift (Stracke et al., 2025) to eliminate the impacts of hyperparameters and deliver high-quality diffusion priors, effectively enhancing the preservation of detailed image information. We formu-

late the extraction of diffusion priors $F_{ref_{sd}} = Enc_{sd}(Img_{ref})$ and $F_{test_{sd}} = Enc_{sd}(Img_{test})$ separately from the reference image $Img_{ref}$ and test image $Img_{test}$, where $Enc_{sd}$ is the CleanDift image encoder.

**Mutil-view cues**. Motivated by the insight that resulting confidence maps remain consistent and robust under rigid geometric transformations of the reference image (Wang et al., 2025), we incorporate multi-view image augmentation. Specifically, we apply random rotations and flips to the images, effectively mitigating the adverse effects of target position and shape variations on feature matching performance. In detail, we formulate the multi-view visual augmentation as the following Eq. 1:

$$Img_{ref_{var_i}} = MultiViewAug(Img_{ref}, M_{ref}) \tag{1}$$

where $M_{ref}$ is the mask of the reference image, and $Img_{ref_{var_i}}$ denotes the reference image after applying the $i_{th}$ image augmentation operation $MultiViewAug(\cdot)$. Available augmentations include one null operation $Identity$, as well as two spatial transformations: random rotation, $RandRotate$, and horizontal flip, $HorizontalFlip$. Afterwards, we perform feature extraction from these transformed reference images $\{Img_{ref_{var_i}}\}_{i=1}^3$. Regarding the $i$th image, we denote its feature as $F_{ref_{var_i}}$. The resulting features are collected into a feature list, termed $F_{ref_{mv}} = \{F_{ref_{var_i}}\}_{i=1}^3$. As for the test image, we only extract its feature $F_{test_{mv}}$ from the original view without peforming image augmentation.

**Visual localization**. To utilize diffusion priors and multi-view cues for localizing regions with the expected semantics, we perform feature matching between features from $Img_{ref}$ and $Img_{test}$. Following existing methods (Tang et al., 2024; Zhang et al., 2024a), the consine similarity in the feature space is calculated. Subsequently, for $F_{ref_{sd}}$ and each element in $F_{ref_{mv}}$, we calculate their similarities with the corresponding test features $F_{test} = \{F_{test_{sd}}, F_{test_{mv}}\}$ respectively. The formulation of feature matching is presented in Eq. 2:

$$S_{vis} = F_{ref} \cdot F_{test}^T \tag{2}$$

where $F_{ref}$ can represent $F_{ref_{sd}}$ and each item in $F_{ref_{mv}}$, and $S_{vis}$ are collections of visual similarity maps. Moreover, the manner in which multi-source visual cues are utilized significantly affects the effectiveness of target matching and localization. Existing methods commonly fuse similarity maps by simple element-wise addition (Tang et al., 2024), which often introduces artifacts outside the target regions. To address the issue, we propose avoiding direct fusion of the confidence maps of feature similarities. Instead, we defer their integration to the generation stage of point prompts, employing an iteratively constraint-based mechanism to guide the point generation process.

## 2.2 TEXT-DRIVEN LOCALIZATION FROM GENERATED PSEUDO-LABELS

Text prompts possess certain advantages in describing common concepts and exhibit stronger capabilities in retrieving targets based on semantic cues within cluttered environments (Rosi & Cermelli, 2025). To exploit the complementarity between modalities, this work proposes an intuitive, seamlessly integrated, and robust text-based localization module to enhance accuracy.

The proposed method is driven by a pseudo-class label list generated by the pretrained image recognition model RAM (Zhang et al., 2024b). As indicated in Eq. 3, we first obtain the predicted label lists $\{Classes_{pseudo_{var_i}}\}_{i=1}^3$ and $Classes_{pseudo_{test}}$, respectively.

$$\begin{aligned} \{Classes_{pseudo_{var_i}}\}_{i=1}^3 &= \{\text{RAM}(Img_{ref_{var_i}})\}_{i=1}^3, \\ Classes_{pseudo_{test}} &= \text{RAM}(Img_{test}), \end{aligned} \tag{3}$$

Based on the group of calculated label lists, an iterative process is performed to determine the most representative pseudo-category labels $Classes_{pseudo}$ via intersection operations as shown in Fig. 3(b), and we formulate the process in Eq. 5,

$$Classes_{pseudo} = \begin{cases} Classes_{pseudo} \cap Classes_{candidate}, & \text{if} \quad Classes_{pseudo} \cap Classes_{candidate} \\ Classes_{pseudo}, & \text{otherwise} \end{cases} \tag{4}$$

where $Classes_{pseudo}$ is initialized as the pseudo-category labels of $I_{ref_{var_0}}$ and $Classes_{candidate}$ represents each element in the iteration. Based on the reasonable assumption that the text label achieving the best semantic alignment across multi-view observations of the same target should be

consistent, the frequently occurring categories in the RAM inference results are selected to form candidate label lists. In cases where multiple labels exist, CLIP (Radford et al., 2021) is employed to compute probability scores and identify the label with the highest probability corresponding to the masked region in the reference image, as indicated by Eq. 5.

$$Class_{pseudo} = \arg \max_{c \in L_{\text{ref}}} \text{CLIP}(Classes_{pseudo}, I_{\text{ref}_{\text{var}_0}})$$ (5)

We observed that CLIP predicted similarity maps corrupted by noise. Instead, CLIP Surgery (Li et al., 2025) is introduced to yield a cleaner textual similarity map $S_{text}$ for improved localization.

Moreover, we select point prompts by iteratively fusing paired similarities $(S_{vis/text}^{pos}, S_{vis/text}^{neg})$. In the $i$-th iteration, a pair of binarized maps, $FG_i$ and $BG_i$, are generated to differentiate foreground and background via Eq. 6,

$$FG_i = \mathbb{I}(S_i^{pos} > weight \cdot S_i^{neg})$$
$$BG_i = \sim FG_i$$ (6)

where $\mathbb{I}(\cdot) = \{(i,j) \mid I(i,j) = 1\}$ extracts the set of foreground pixels. $FG_i$ is adaptively fused with $FG_{i-1}$ from the previous iteration as indicated by Eq. 7,

$$FG_i = \begin{cases} FG_{i-1} \wedge FG_i, & \text{if } |FG_{i-1} \wedge FG_i| \geq \tau \cdot |FG_{i-1}| \\ FG_{i-1}, & \text{otherwise} \end{cases}$$ (7)

where $\tau$ is the threshold for the ratio of overlapped foreground pixels between consecutive iterations, and $|\cdot|$ measures the non-zero pixel count. By iterating on paired similarities, multi-modal prompts are fully exploited to refine the target semantic region progressively. Afterwards, we apply K-Means clustering to the final indicators to generate positive prompts $\{P_m^{pos}\}_{m=1}^{M}$ and negative prompts $\{P_n^{neg}\}_{n=1}^{N}$. $M$ and $N$ are the predefined number of cluster centers.

## 2.3 Consensus-Oriented Mask Proposer

Excessively dense positive point prompts for SAM lead to over-segmentation, whereas inappropriate negative points may cause under-segmentation. To address this, we propose a mask proposal generator, COMP, based on the consensus principle, which reframes mask filtering and refinement as a proposal voting problem. It consists of a self-correcting process with four adaptive steps: reconfirming, voting, outlier removal, and inlier repainting, shown in Fig. 3(c). They can be treated as multiple experts negotiating to reach a consensus on the final mask proposals. The first two procedures act as neutral experts, outlier removal as the radical expert, and inlier repainting as the conservative expert, collaboratively refining mask proposals. Specifically, for initial mask proposals ($MP_{ini}$ derived from previously generated point prompts), COMP leverages the interaction between four experts to ensure the generation of high-quality mask proposals $MP_{comp}$. Subsequently, these proposals are sent back to SAM to obtain the final segmentation result $M_{output}$.

**Reconfirming**. We reevaluate the necessity of each proposal in $MP_{ini}$ by thresholding their similarities with both the referenced region $M_{ref} \circ F_{ref_{var_0}}$ and the generated pseudo label $Class_{pseudo}$. The recalculated similarities are represented as $RS_{mask}$ and $RS_{text}$. Afterwards, using two predefined thresholds, $T_{mask}$ and $T_{text}$, along with the TopK operator, we sequentially filter out the resulting candidate proposals $MP_{reconfirm}$ with high similarity ratings. In this paper, $T_{mask}$ and $T_{text}$ are empirically set to 50 and 20, respectively.

**Voting**. Upon obtaining filtered mask proposals $MP_{reconfirm}$ in the first stage, we adopt another neutral strategy to merge existing masks via a voting mechanism. In the implementation, we splat all mask proposals onto the same two-dimensional plane and aggregate them through element-wise addition. For each location, all pixels with votes exceeding a certain proportion, $T_{voting}$, of the total pixel number are selected to form the merged binary mask proposal $MP_{voting}$.

**Outlier Removal**. There are potential outliers that fail to be removed from the merged mask proposal $MP_{vote}$ in the preceding stages. Exploiting the property that the spatial distribution of object masks tends to be continuous and smooth in their local regions, we propose to remove outlier pixels in an aggressive manner using morphological erosion operations, which effectively eliminate isolated noise pixels. The resulting mask proposal is defined as $MP_{erode}$.

**Inlier Repainting**. This operation serves to "deny the denied" and "smooth the unsmoothed". The former restores mask regions that were mistakenly removed by reintegrating them back to the mask

Table 1: Quantitative results on the PerSeg dataset. The average mIoU and scores of each representative categories are presented. The best methods are indicated by **underlined bold** text, while the second-ranking and third-ranking methods are emphasized in **bold** and underlined, respectively.

| | Venue | Mean | Backpack | Barn | Can | Cat | Clock | Robot Toy | Teddy Bear | Thin Bird |
|---|---|---|---|---|---|---|---|---|---|---|
| *Training* | | | | | | | | | | |
| VP | NIPS 2022 | 65.9 | 66.7 | 58.6 | 61.2 | 76.6 | 59.2 | 72.4 | 79.8 | 67.4 |
| Painter | CVPR 2023 | 56.4 | 88.1 | 3.2 | 19.1 | **94.1** | 42.9 | 65.0 | 93.0 | 20.9 |
| SEEM | NIPS 2023 | 87.1 | 94.1 | 82.5 | 65.4 | 91.1 | 72.4 | 95.8 | **95.2** | 71.3 |
| SegGPT | ICCV 2023 | **94.3** | 94.4 | 63.8 | 96.6 | **94.1** | 92.6 | **96.2** | 93.7 | 92.6 |
| *Training-Free* | | | | | | | | | | |
| PerSAM | ICLR 2024 | 89.3 | 95.4 | 38.9 | 96.2 | **94.1** | **96.2** | 60.6 | 94.6 | **93.7** |
| Matcher | ICLR 2024 | 94.1 | 95.6 | **94.6** | 96.1 | 93.4 | 91.3 | 95.2 | 94.8 | 90.8 |
| IPSeg | IJCV 2025 | 90.9 | **96.3** | 93.5 | 80.9 | **94.1** | 73.2 | 65.8 | 86.8 | 93.1 |
| Ours | - | **95.1** | 95.9 | **96.4** | **97.3** | **95.2** | 94.8 | **96.5** | **95.9** | **93.5** |

Table 2: Quantitative results on the COCO-20$^i$ and FSS datasets. Results of all folds are provided for the COCO-20$^i$ dataset. The top three are represented in the same way as in Table 1.

| | Venue | COCO | | | | | FSS |
|---|---|---|---|---|---|---|---|
| | | Fold0 | Fold1 | Fold2 | Fold3 | mean | mean |
| *Training* | | | | | | | |
| HSNet | ICCV 2021 | 37.2 | 44.1 | 42.4 | 41.3 | 41.2 | 86.5 |
| VAT | ECCV 2022 | 39.0 | 43.8 | 42.6 | 39.7 | 41.3 | **90.3** |
| FPTrans | NIPS 2022 | 44.4 | 48.9 | 50.6 | 44.0 | 47.0 | - |
| MSANet | arXiv 2022 | 47.8 | **57.4** | 48.7 | 50.5 | 51.1 | - |
| Painter | CVPR 2023 | 31.2 | 35.3 | 33.5 | 32.4 | 33.1 | 61.7 |
| SegGPT | ICCV 2023 | **56.3** | **57.4** | **58.9** | **51.7** | **56.1** | 85.6 |
| *Training-Free* | | | | | | | |
| PerSAM | ICLR 2024 | 23.1 | 23.6 | 22.0 | 23.4 | 23.0 | 71.2 |
| Matcher | ICLR 2024 | 52.2 | 53.3 | 52.5 | **51.7** | 52.4 | 87.0 |
| Ours | - | **52.7** | 53.7 | **52.6** | 52.1 | **52.8** | **87.4** |

proposal, while the latter fills gaps within the mask to ensure spatial continuity. Morphological dilation operations are applied to detect pixels wrongly excluded and to identify internal discontinuities, enabling their correction.

## 3 EXPERIMENTS

To validate the superiority of MMSeg, we select several representative reference-based methods and specialized few-shot segmentation models for comparative experiments. MMSeg requires no dataset training and operates solely by leveraging pretrained Vision Foundation Models for segmentation based on reference images and masks. We utilize DINOv2 as a general visual feature extractor and CleanDIFT as a low-level visual feature extractor (Tang et al., 2024). RAM and CLIP-Surgery are used to identify and generate pseudo-category labels, while SAM functions as a class-agnostic image segmentation model. More details about dataset and evaluation metric are in Appendix.

### 3.1 QUANTITATIVE EXPERIMENTS

We display results on the PerSeg dataset in Table 1. Our method outperforms both training-based and training-free baselines. Concretely, it surpasses the second-best method, SegGPT, by 0.8% and outperforms Painter by a substantial margin of 38.7%. As for per-category analyses, while PerSAM exceeds our method on a few categories, it suffers from unstable segmentation performance, with considerably low IoU scores on the Barn and Robot Toy classes. However, our results are well-balanced and consistently high, with IoU scores exceeding 90% for most categories. The robustness and high accuracy stem from our core designs: visual localization and text-driven localization. These strategies enrich feature representation and enable more precise segmentation, effectively equipping our method to handle diverse personalized visual concepts.

Experimental results on the $COCO$-$20^i$ and FSS datasets are summarized in Table 2. Our method also achieves leading performance. Compared to other training-free baselines, MMSeg delivers

Table 3: Quantitative results on the part segmentation datasets, including PASCAL-Part and PACO-Part. The per-class specifics are provided for the former, while the per-fold results are displayed for the latter. The top three are represented in the same way as in Table 1.

| | PASCAL-Part | | | | | PACO-Part | | | | |
| | animals | indoor | person | vehicles | mean | F0 | F1 | F2 | F3 | mean |
|---|---|---|---|---|---|---|---|---|---|---|
| *Training* | | | | | | | | | | |
| HSNet | 21.2 | 53.0 | 20.2 | 35.1 | 32.4 | 20.8 | 21.3 | 25.5 | 22.6 | 22.6 |
| VAT | 21.5 | 55.9 | 20.7 | 36.1 | 33.6 | 22.0 | 22.9 | 26.0 | 23.1 | 23.5 |
| Painter | 20.2 | 49.5 | 17.6 | 34.4 | 30.4 | 13.7 | 12.5 | 15.0 | 15.1 | 14.1 |
| SegGPT | 22.8 | 50.9 | 31.3 | 38.0 | 35.8 | 13.9 | 12.6 | 14.8 | 12.7 | 13.5 |
| *Training-Free* | | | | | | | | | | |
| PerSAM | 19.9 | 51.8 | 18.6 | 32.0 | 30.1 | 19.4 | 20.5 | 23.8 | 21.2 | 21.2 |
| Matcher | **37.1** | 56.3 | **32.4** | 45.7 | 42.9 | **32.7** | **35.6** | 36.5 | **34.1** | **34.7** |
| ours | **35.3** | **64.5** | 32.3 | **46.8** | **44.7** | **31.2** | **34.3** | 38.6 | **33.5** | **34.4** |

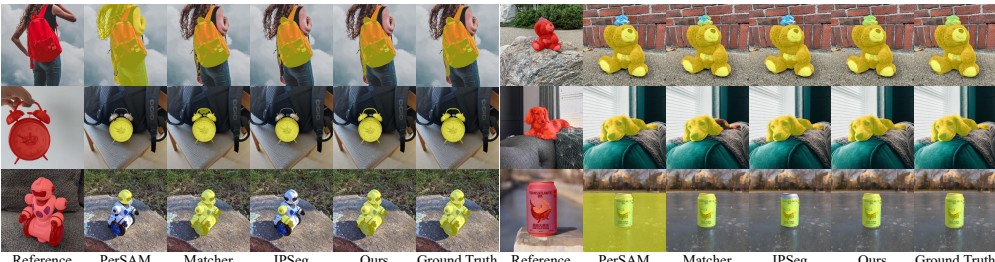

Figure 4: Qualitative results on PerSeg. In each half of the figure, the leftmost column contains reference images and their masks for each row. The remaining columns visualize segmentation results from PerSAM, Matcher, IPSeg, MMSeg, and the Ground Truth.

substantial advantages, surpassing PerSAM by 29.8% and 16.2% on both datasets, respectively. Although SegGPT has been trained on in-domain data similar to these datasets, our method defeats SegGPT on $COCO$-$20^i$-fold3 and outperforms it by 1.8% on the FSS dataset. These results demonstrate that our approach achieves superior segmentation performance on unseen data. Furthermore, we also include comparisons with specialized baselines for few-shot segmentation. Our method excels over MSANet by 1.7% and HSNet by 11.6% on the $COCO$-$20^i$ dataset.

Furthermore, the results on the fine-grained part segmentation datasets are presented in Table 3. On the PASCAL-Part dataset, MMSeg shows a significant improvement of 1.8% over the next-best method and 14.6% over PerSAM. On the PACO-Part dataset, it also achieves top-tier performance comparable to Matcher and significantly outperforms other peer methods. These results underscore the strong generalization of MMSeg to segmenting targets across diverse semantic scales, categories, and sizes without task-specific training.

## 3.2 QUALITATIVE EXPERIMENTS

Fig. 4 shows qualitative comparisons on the PerSeg dataset. Utilizing the aligned text-image feature space of CLIP, our method employs generated pseudo-labels to accurately identify segmentation targets and exclude extraneous elements, such as the person in the first image. The integration of multi-view reference images further enriches feature extraction. In the second row, while PerSAM and IPSeg only partially segment the alarm clock and dog, our method achieves complete segmentation. Notably, our approach captures finer details, such as backpack straps and teddy bear hat edges, which is attributed to our mask proposal strategy that discards unsuitable masks while merging suitable ones, resulting in accurate and coherent segmentations.

We visualized comparison results on the FSS dataset in Fig. 5. The fourth and fifth images in the left panel illustrate our method's effective control over segmentation granularity. The final row confirms our ability to distinguish regions with similar semantics, while competitors struggle with unrelated content; for instance, Matcher missegments the background, and PerSAM fails to segment the target. The introduction of pseudo-labels enhances semantic understanding, enabling complete target segmentation and avoiding issues like recognizing only part of a boxing glove or pump. The

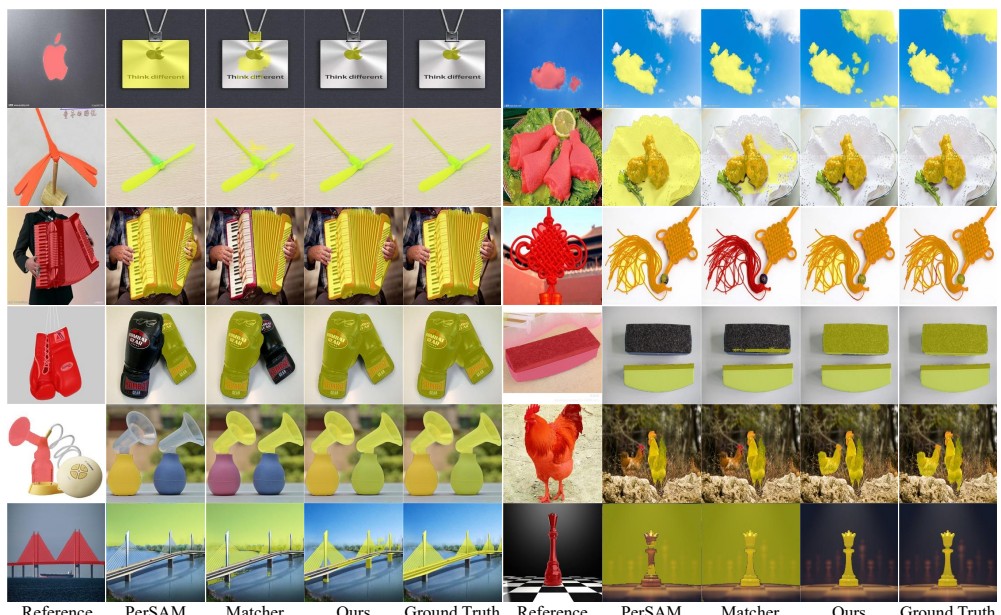

Figure 5: Qualitative results on FSS. The layout is similar to Fig. 4. The results from PerSAM, Matcher, and our method are visualized.

first three images in the left panel showcase MMSeg's robustness against background interference, thanks to its text-driven target localization and mask proposal refinement. During the reconfirmation stage of COMP, we effectively eliminate irrelevant mask proposals, ensuring final mask quality.

### 3.3 ABLATION EXPERIMENTS

In this section, we conduct comprehensive ablation studies to validate the contribution of each component. As detailed in Table 4, the results systematically demonstrate that each module yields significant mIoU gains.

**Effect of VLDM**. This module exploits enriched features from diffusion priors and multi-view augmented reference images for precise visual localization. A comparison between the first two rows reveals that the introduction of VLDM improves performance by 1.68% and

Table 4: Quantitative ablation results on the PerSeg and FSS datasets. The "Gain" column indicates the relative improvement of each ablation group relative to the baseline.

|  | PerSeg | | FSS | |
|---|---|---|---|---|
|  | mIoU | Gain | mIoU | Gain |
| baseline | 91.61 | - | 78.16 | - |
| + VLDM | 93.29 | 1.68 | 85.78 | 7.62 |
| + TLGP | 92.37 | 0.76 | 85.3 | 7.14 |
| + VLDM + TLGP | 92.67 | 1.06 | 86.5 | 8.34 |
| + VLDM + COMP | 94.42 | 2.81 | 86.82 | 8.66 |
| + TLGP + COMP | 93.67 | 2.06 | 86.43 | 8.27 |
| MMSeg | 95.1 | 3.49 | 87.4 | 9.24 |

7.6% on the PerSeg and FSS datasets, respectively. When integrated into the full model in the last two rows, it still delivers notable improvements of 1.4% and 1.0% on these datasets.

**Effect of TLGP**. This module provides essential semantic context, enhancing performance beyond visual features alone. The comparison of the first and third rows reveals a significant 7.1% improvement on the FSS dataset with TLGP integration. Additionally, comparisons between the fifth and seventh rows show accuracy gains of 0.7% and 0.6% on the PerSeg and FSS datasets, respectively, when combined with other components. These results underscore the importance of semantic guidance from generated pseudo-labels. For clarity, we visualize similarity maps in Fig. 6, illustrating the relationship between high-level visual features from the reference image, the test image, and the pseudo-label. The text-based similarity map enables more holistic target localization, effectively reducing granularity misalignment and background interference.

**Effect of COMP**. This pipeline consistently enhances mask quality, achieving mIoU gains of 0.7% to 2.4% across datasets and model configurations. Its effectiveness arises from a self-correction mechanism and multi-aspect generation constraints that refine mask proposals. Fig. 7 illustrates the predicted masks before and after incorporating COMP, highlighting significant flaws in masks

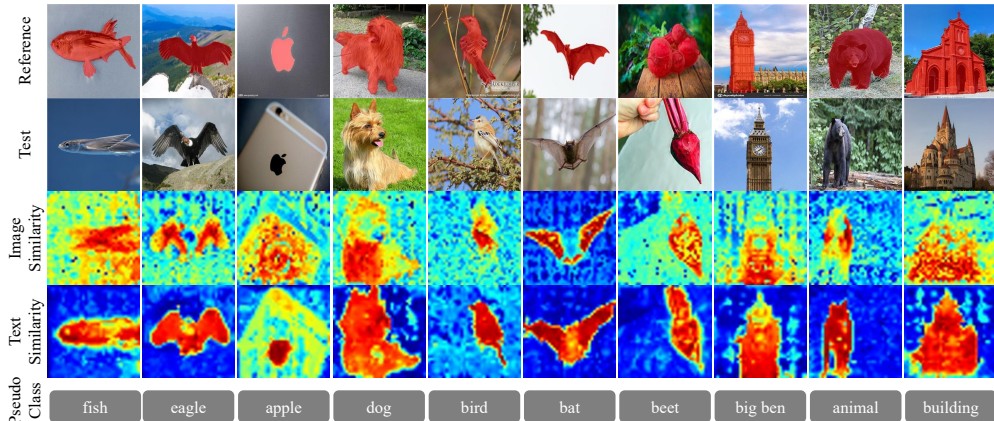

Figure 6: Visualization of feature similarity maps. The first and second rows are reference images and test images. The third and fourth rows displays similarities between high-level visual features or visual-text features. The last row provides generated pseudo-labels for each case.

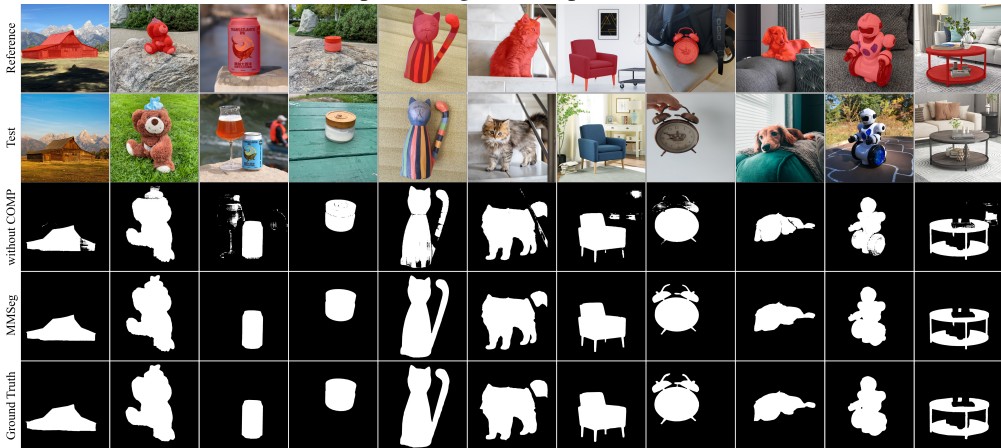

Figure 7: Qualitative ablation results on COMP. The top two rows are reference and test images. The third and fourth rows compare the mask quality before and after incorporating COMP. In the last row, the corresponding ground truth are provided for reference.

without COMP, such as localized omissions, peripheral outliers, and central-region discontinuities. These observations underscore COMP's role in optimizing mask quality.

In summary, the ablation studies validate that the visual and textual localization branches (VLDM and TLGP) provide complementary feature enhancements, while COMP serves as a critical final refinement stage. Together, they achieve a synergistic leap in segmentation performance, building on the quantitative mIoU gains observed earlier.

## 4 CONCLUSION

This paper proposes MMSeg, multi-modal and multi-view driven semantic enrichment for training-free image prompt segmentation. MMSeg consists of two key components: object localization and mask generation, enhancing reference feature representation through multi-modal prompts and visual cues from multi-view image augmentation. For localization, we boost feature discriminability via diffusion model priors and ensure robustness by maintaining segmentation consistency across multi-view references. To harness multi-modal complementarity, we integrate text modality and generate pseudo-category labels for semantic guidance, forging synergy between visual and textual cues. For mask generation, we design a self-correcting pipeline that aggregates consensus across multiple proposals to optimize mask precision and fidelity. Extensive experiments on benchmark datasets demonstrate that MMSeg achieves competitive performance in both semantic and part segmentation tasks, with notable improvements in segmentation quality.

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

# A APPENDIX

## A.1 REPRODUCIBILITY STATEMENT

We have already elaborated on all the models or algorithms proposed, experimental configurations, and benchmarks used in the experiments in the main body or appendix of this paper. Furthermore, we declare that the entire code used in this work will be released after acceptance.

## A.2 THE USE OF LARGE LANGUAGE MODELS

We use large language models solely for polishing our writing, and we have conducted a careful check, taking full responsibility for all content in this work.

## A.3 RELATED WORK

### A.3.1 VISION FOUNDATION MODELS

Vision foundation models (VFMs) have demonstrated remarkable generalizability across a wide range of downstream tasks. The CLIP families (Radford et al., 2021; Kolesnikov et al., 2023; Sun et al., 2024b) perform contrastive learning on text-image pairs to construct a unified text-image feature space, enabling open-world perception. SAM and SAMv2 figure out segmentation by training on large-scale image and video datasets, supporting domain-agnostic segmentation via visual prompting. SEEM further extends this capability by incorporating multi-modal prompts and interactive refinement. Through self-supervised pretraining at scale, DINO and DINOv2 extract highly generalizable and semantically rich visual features that facilitate various applications, including semantic correspondence and depth estimation (Caron et al., 2021; Oquab et al., 2024). DINOv3 advances these capabilities with improved feature discrimination and enhanced domain generalization (Siméoni et al., 2025). Recently, researchers have begun leveraging internal feature representations from pretrained diffusion models (Rombach et al., 2022). However, existing methods often require meticulous tuning of layer selection and iteration counts, resulting in unstable feature quality and compromised segmentation performance. To mitigate these limitations, CleanDIFT (Stracke et al., 2025) introduces a feature enhancement strategy that extracts more stable and higher-quality diffusion features without extensive tuning.

### A.3.2 REFERENCE-BASED SEGMENTATION

Reference-based segmentation methods can be categorized by reference modality. Text-referenced methods, such as Grounded SAM (Ren et al., 2024b) and DINO-x (Ren et al., 2024a), segment regions aligned with natural language descriptions. Visual-referenced approaches, including SAM (Kirillov et al., 2023), Painter (Wang et al., 2023a), and SegGPT (Wang et al., 2023b), use visual prompts to segment regions of interest, while Dinov (Li et al., 2024) introduces flexible visual context prompts supporting multiple reference images. Multi-modal-referenced segmentation integrates both text and visual cues, as seen in SEEM (Zou et al., 2023) and VLP-SAM (Sakurai et al., 2025), which require full training or fine-tuning. SoT benchmark Rosi & Cermelli (2025) reveals that text prompts perform better in cluttered environments, while visual prompts capture complex concepts more effectively. To the best of our knowledge, no existing method efficiently combines both modalities in a training-free framework. To bridge this gap, we propose a novel multi-modal prompting framework for segmentation that utilizes only image prompts. Our method leverages pretrained Vision Foundation Models (VFMs) by employing multi-view image augmentation and generating pseudo-category labels, all without the need for additional parameters or training.

### A.3.3 AUTOMATIC PROMPTING FOR SAM

The practical deployment of SAM requires domain expertise for manual prompt design and iterative refinement, particularly in specialized fields such as medical imaging and remote sensing. High-quality guidance of the segmentation process requires specific knowledge, such as identifying pathological tissues or interpreting spectral signatures. This reliance on skilled operators incurs significant time and training costs. Automated prompt engineering seeks to alleviate these limita-

tions by generating optimal prompts for SAM, reducing user burden while enhancing adaptability to complex visual domains (Espinosa et al., 2025).

Existing automated prompt engineering methods for SAM can be classified into two categories. Learning-based approaches enable SAM to generate prompts from textual, visual, or other reference modalities by fine-tuning prompt encoders or introducing additional parameters. For instance, VRP-SAM (Sun et al., 2024a) trains a visual prompt encoder to produce embeddings from reference images, whereas FM-PPO (Liu et al., 2025a) learns a policy function for direct prompt point selection without manual threshold configuration. In specialized domains, expert knowledge can substantially enhance the prompt generation process. The PointPrompt benchmark (Quesada et al., 2024) consolidates human-annotated trajectory data to evaluate existing methods. SegAgent (Zhu et al., 2025) further fine-tunes a multi-modal large language model to emulate human annotators, automatically generating suitable point prompts for SAM. However, these learning-based approaches incur huge computational costs, limiting their accessibility for users with constrained resources.

Training-free methods utilize predefined rules to select candidate prompts. SAMAug (Dai et al., 2023) systematically evaluates point-based prompting strategies, providing insights for future research. PerSAM (Zhang et al., 2024a) designates the most and least similar pixels as positive and negative prompts, but often results in clustered distributions in complex scenes. IPSeg (Tang et al., 2024) and SuperPromptSeg (Zhou et al., 2025) enhance prompt efficiency through clustering techniques. IPSeg selects top-k cluster centers, while SuperPromptSeg uses superpixel-based feature clustering. Among training-free methods, Matcher achieves notable performance improvements via bidirectional matching and multi-granularity constraints, albeit with considerable computational overhead and sensitivity to hyperparameters. GBMSeg (Liu et al., 2024a) further refines this paradigm with three sampling strategies to retain high-value prompts while minimizing spatial concentration and positive-prompt bias. Synpo (Liu et al., 2025b) innovates by selecting negative prompts from semantically ambiguous regions rather than merely dissimilar backgrounds. In contrast, our approach introduces adaptive target localization combined with mask self-correction in a fully training-free pipeline, simplifying the process by avoiding complex criteria and human annotations.

### A.4 DATASET AND EVALUATION METRIC

We conduct experiments on the PerSeg (Zhang et al., 2024a), FSS (Li et al., 2020), and COCO-$20^i$ (Nguyen & Todorovic, 2019) dataset. The PerSeg benchmark is specifically designed for segmentation methods using image prompts. It covers 40 diverse object categories, with a total of 216 images for testing (5 to 7 per category). The COCO-$20^i$ dataset (Nguyen & Todorovic, 2019) is a subset of COCO (Lin et al., 2014), partitioned using a four-fold cross-validation split. Each fold contains 20 categories and 1,000 reference-test image pairs. The FSS dataset (Li et al., 2020) is a large-scale dataset for few-shot segmentation tasks, comprising 1,000 categories: 520 for training, 240 for validation, and 240 for testing, with 10 images per category. Our method performs inference directly on the full PerSeg dataset, the COCO-$20^i$ validation categories, and the FSS test categories without any training. Furthermore, we introduce part segmentation datasets to validate the effectiveness across varying semantic granularities. The PASCAL-Part (Chen et al., 2014) dataset includes part-level annotations for animals, indoor objects, persons, and vehicles, with 13 categories in total. The PACO dataset (Ramanathan et al., 2023) provides higher semantic granularity and diversity, with 75 object classes and over 450 part classes.

For all datasets, we use mean Intersection over Union (mIoU) as the evaluation metric. Regarding COCO-$20^i$, we follow the same evaluation protocol in Matcher, reporting results for each fold and the average performance across all folds.

### A.5 MORE VISUALIZATION OF EXPERIMENTAL RESULTS

In this subsection, we present additional visualization experimental results. Figure 8 illustrates the visualization of TLGP ablation experiments, while Figure 9 depicts the visualization of COMP ablation experiments. Figures 10 and 11 showcase the visualizations of comparative experiments conducted on the FSS and PerSeg datasets, respectively.

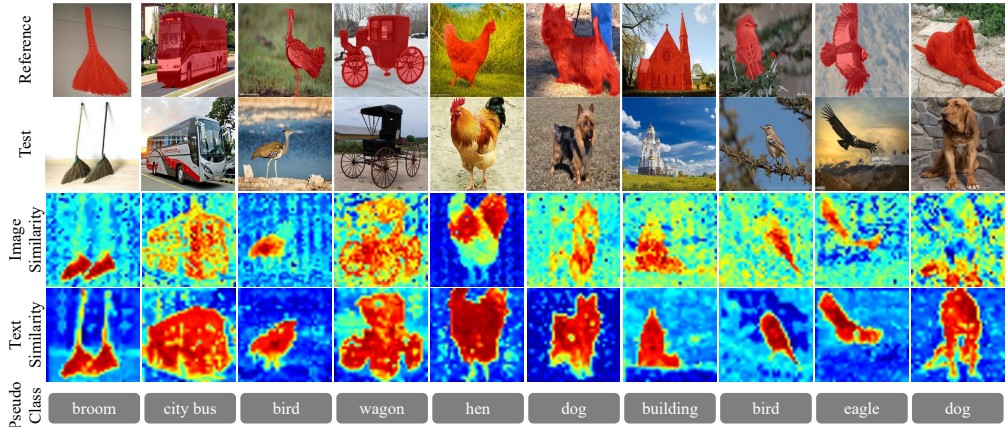

Figure 8: More visualization of feature similarity maps. The first and second rows are reference images and test images. The third and fourth rows displays similarities between high-level visual features or visual-text features. The last row provides generated pseudo-labels for each case.

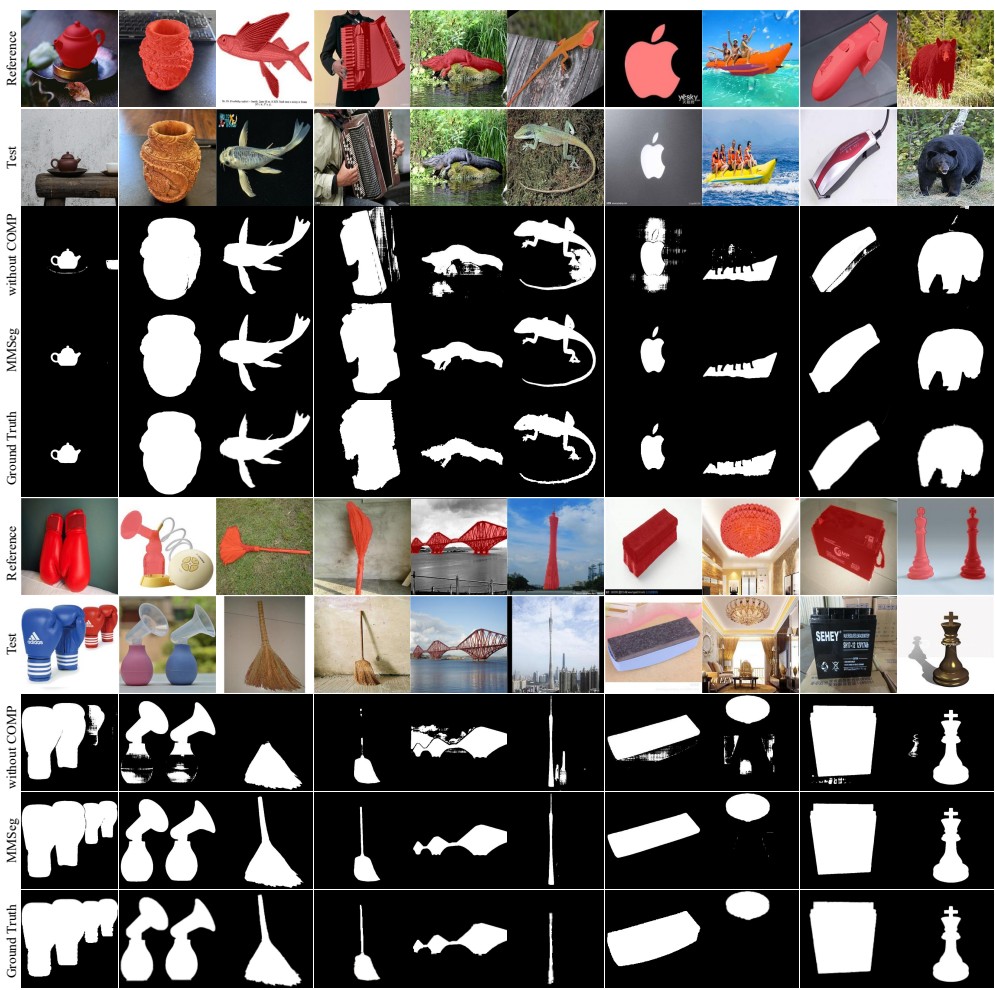

Figure 9: More qualitative ablation results on COMP. The figure can be divided into two sections: the upper and lower parts. In each part, the top two rows are reference and test images. The third and fourth rows compare the mask quality before and after incorporating COMP. In the last row, the corresponding ground truth is provided for reference.

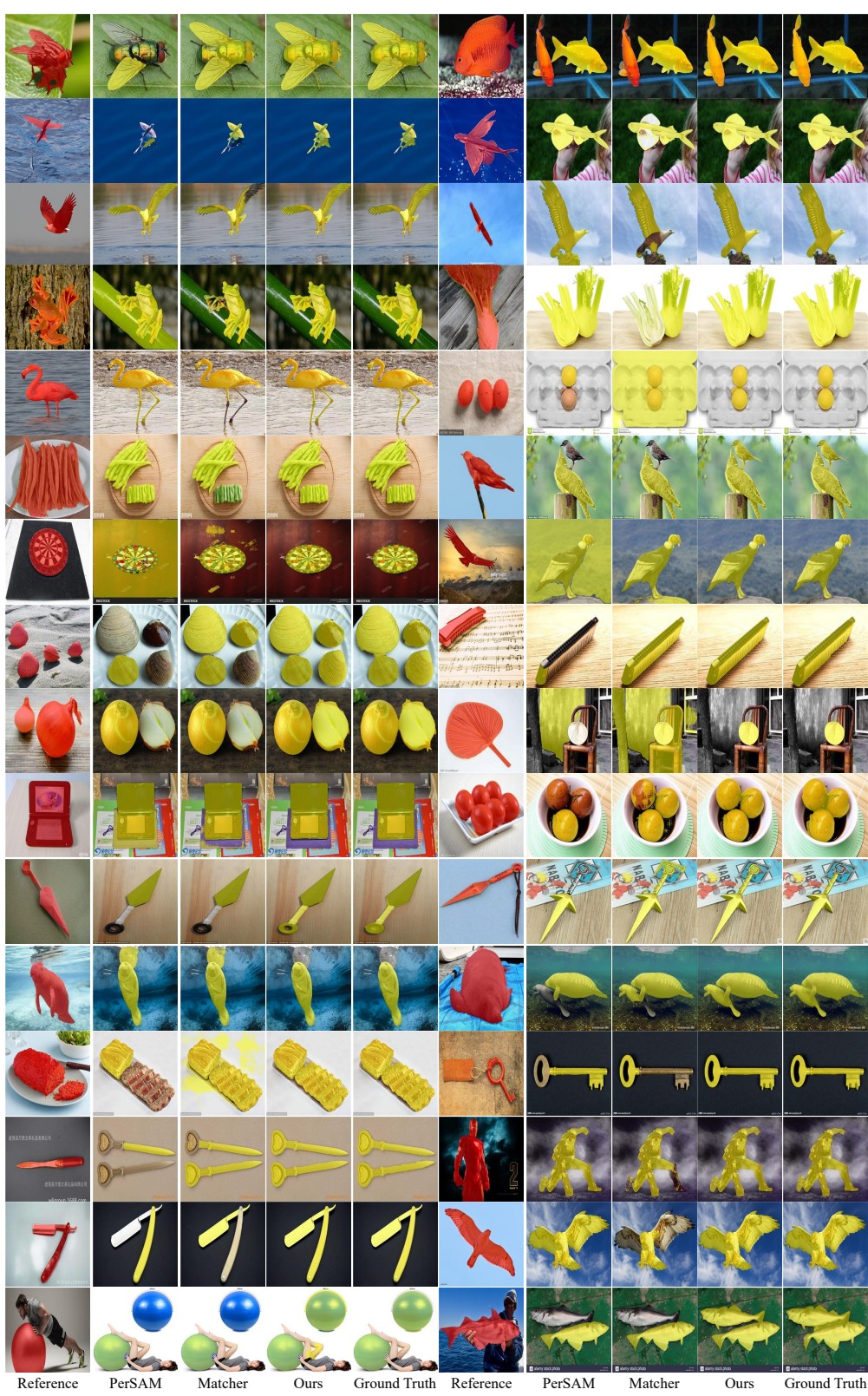

Figure 10: More qualitative results on FSS. In each half of the figure, the leftmost column contains reference images and their masks for each row. The remaining columns visualize segmentation results from PerSAM, Matcher, MMSeg, and the Ground Truth.

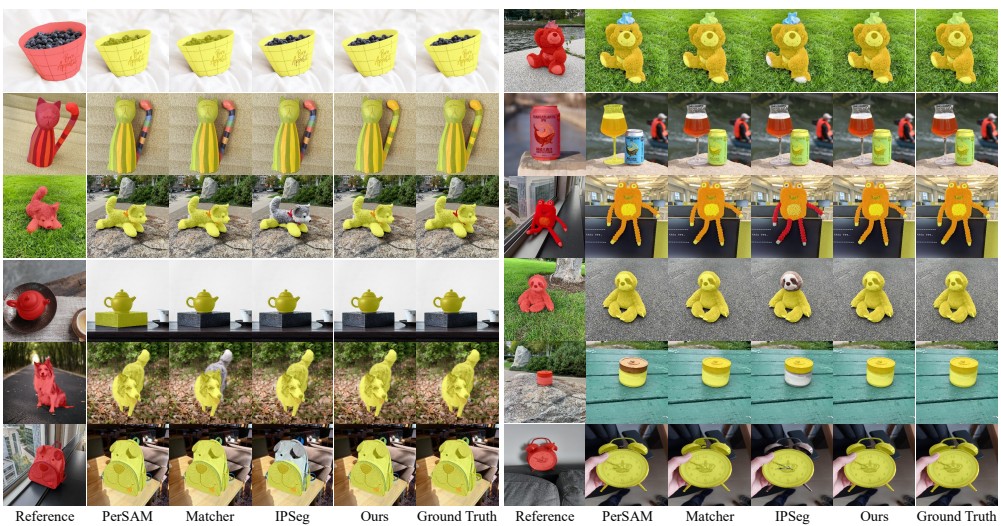

Reference   PerSAM   Matcher   IPSeg   Ours   Ground Truth    Reference   PerSAM   Matcher   IPSeg   Ours   Ground Truth

Figure 11: More qualitative results on PerSeg. The layout is similar to Fig. 10. The results from PerSAM, Matcher, IPSeg, and our method are visualized.

