# OpenReview forum: "MMSeg: Multi-Modal and Multi-View Driven Semantic Enrichment for Training-Free Image Prompt Segmentation"
_ICLR.cc/2026/Conference — ICLR 2026 Conference Withdrawn Submission_

### Official Review · Reviewer_pLq6 · 2025-10-27

**Soundness:** 3
**Presentation:** 1
**Contribution:** 2
**Rating:** 2
**Confidence:** 5

**Summary:**

The authors propose MMSeg, a multi-modal and multi-view framework to enhance semantic representation by integrating two localization modules: 1) a visual module (VLDM) augmented with diffusion priors and multi-view cues, and 2) a text-driven module (TLGP) based on generated pseudo-labels. These components are designed to provide complementary information for more precise localization. Finally, a consensus-oriented mask proposer (COMP) is employed to filter and refine the segmentation output.

**Strengths:**

- The paper proposes a novel training-free MMSeg, which strategically integrates multi-modal (visual and pseudo-text) and multi-view cues. This approach thoughtfully addresses the semantic representation limitations of current single-modality image-prompt methods.
- The introduction of two complementary localization modules (VLDM and TLGP) is a significant contribution. This design leverages diffusion priors, multi-view consistency, and pseudo-label generation to effectively mitigate the localization inaccuracies common in methods that rely solely on high-level feature matching.
- The inclusion of a consensus-oriented mask proposer (COMP) provides a structured mechanism to filter and refine mask proposals, directly addressing the common challenge of poor mask quality in prompt-based segmentation.

**Weaknesses:**

1. The paper focuses on segmentation based on a single reference image, which is a limited setting. In real-world scenarios, many concepts cannot be fully expressed by a single image prompt, and the proposed framework would likely fail in such cases. More general forms of visual context should be considered.
1. The description regarding "low-level" and "high-level" features between Lines 150 and 155 is confusing. The authors first state in Line 150 that most methods ignore the "crucial role" of "low-level and diverse features". However, they then mention in Line 152 using "high-level image features" to enhance image features, only to state again in Line 155 that the branch integrates "low-level feature." This inconsistent terminology is confusing.
1. During the extraction of multi-view cues, is the same image encoder from CleanDift still used? If so, would $F_{test_{sd}}$ and $F_{test_{mv}}$ in $F_{test}$ be identical features? This lacks important details and requires clarification.
1. The paper introduces an excessive number of symbols and notations without clear definitions or distinctions, which significantly hinders readability. This needs improvement. For example, in Equ. 5, what do $c$ and $L_{ref}$ in $\arg\max_{c \in L_{ref}}$ represent? In Equ. 6, what do "weight" and the $\sim$ symbol denote? In Equ. 7, what does the $\wedge$ symbol represent?
1. The structure of Sec. 2.2 is very confusing.
    1. The authors seem to introduce two iterative processes: one for handling class labels and another for point prompts. However, crucial details regarding the motivation, execution order, number of iterations, and results of these processes are missing.
    1. The first half of the section focuses on pseudo-label construction, but after Equ. 5, it abruptly transitions to similarity maps, then directly to the iterative construction of point prompts, and subsequently introduces clustering. The writing lacks necessary transitions and explanations, making the overall organization incoherent and difficult to follow.
1.  In Line 238, the authors use K-Means clustering to generate positive and negative point prompts. Does the choice of this specific clustering algorithm affect the final performance? Additionally, how does the setting of the number of clusters impact the results?
1. The authors have omitted a comparison of inference efficiency between the proposed algorithm and existing methods. Furthermore, the ablation studies lack a performance-efficiency analysis of the different components.
1. The granularity of the ablation study is too coarse. The individual modules (VLDM, TLGP, COMP) involve combinations of multiple components and processes, but the authors have overlooked ablation on these crucial details:
    1. The individual contributions and combined design of the diffusion branch and the multi-view augmentation branch within VLDM.
    2. Details in TLGP, such as the pseudo-label construction method, sensitivity to the RAM model, the concrete design of the two iterative processes, and the impact of the K-Means algorithm's choice and settings.
    3. Details for COMP, including the motivation for its multi-step operations, the rationale of the workflow design, and the justification for hyperparameter choices.

**Questions:**

See Weaknesses.

---

### Official Review · Reviewer_4nsu · 2025-10-30

**Soundness:** 2
**Presentation:** 2
**Contribution:** 2
**Rating:** 4
**Confidence:** 4

**Summary:**

This paper introduces MMSeg, a training-free framework for image-prompted segmentation that leverages multi-modal (visual and textual) and multi-view cues to enhance semantic representation. The method consists of two main stages: (1) Object Localization, which uses a Visual Localization module augmented by Diffusion prior and Multi-view cues (VLDM) and a Text-driven Localization module from Generated Pseudo-labels (TLGP), and (2) Mask Generation, which employs a Consensus-Oriented Mask Proposer (COMP) to refine initial mask proposals from SAM. The core idea is to overcome the limitations of single-modality prompting by enriching features and leveraging segmentation consistency. The authors demonstrate state-of-the-art performance on several benchmarks.

**Strengths:**

1. Training-Free Paradigm. The method's ability to achieve such strong performance without any fine-tuning is a major practical advantage, enhancing its reproducibility and ease of adoption for various applications.
2. Effective Ablation Studies. The ablation study is comprehensive and effectively validates the contribution of each proposed component (VLDM, TLGP, COMP), clearly showing their cumulative benefit.
3. Good Performance. The paper provides extensive experimental validation across multiple datasets (PerSeg, FSS, COCO-20ⁱ, PASCAL-Part, PACO-Part). The results are impressive, showing clear improvements over a wide range of strong baselines, including both training-free and training-based methods. The performance on fine-grained part segmentation is particularly noteworthy.

**Weaknesses:**

1. Methodological Complexity and Efficiency: The pipeline is complex, involving multiple feature extractors (DINOv2, CleanDIFT, RAM, CLIP-Surgery) and a multi-stage processing flow. A discussion or analysis of the computational cost, inference speed, and parameter count compared to other methods is missing. This is crucial for assessing the method's practicality for real-time or resource-constrained applications.
2. Multi-modal Prompting: The idea of combining visual and textual prompts for segmentation is not new. Methods like ViTron, GPT-4V, UniFSS, and even earlier works with CLIP+SAM variants have explored this paradigm. The novelty claimed here is incremental, focusing on a training-free, automatic generation of the text prompt (via pseudo-labels) within a specific pipeline. The paper fails to properly situate itself against this existing landscape of multi-modal segmentation.
3. The "Training-Free" Aspect: The community widely acknowledges that SAM-enabled, training-free segmentation is an active research direction. The novelty, therefore, shifts from being training-free to how effectively one is training-free. The proposed method, while effective, is a complex assemblage of existing models (DINOv2, Diffusion prior, RAM, CLIP-Surgery, SAM). The lack of comparison with SAM 2, which natively supports few-shot segmentation, is a critical omission. It raises the question: does this complex framework still provide a significant advantage over a powerful, unified foundation model like SAM 2 out-of-the-box?
4. Weak and Under-Explained Motivation for Multi-View Cues: The motivation for using multi-view augmentation is critically underdeveloped in the introduction and methodology. The paper states it mitigates the effects of "target position and shape variations," but this is vague. A stronger motivation would be to enforce geometric invariance in the feature representation, ensuring the model matches the object's identity regardless of its pose or orientation in the reference image. This lack of a clear, upfront rationale makes the component feel like a standard engineering trick rather than a principled design choice.

**Questions:**

See the weaknesses.

---

### Official Review · Reviewer_d9dm · 2025-10-31

**Soundness:** 3
**Presentation:** 3
**Contribution:** 2
**Rating:** 4
**Confidence:** 4

**Summary:**

This paper proposes MMSeg, a training-free multi-modal and multi-view segmentation framework that enhances semantic richness by combining diffusion priors, multi-view visual cues, and text-driven pseudo-labels for improved object localization and mask quality. Extensive experiments across multiple benchmarks (PerSeg, FSS, COCO-20i, PACO-Part) demonstrate that MMSeg achieves state-of-the-art performance among training-free segmentation methods.

**Strengths:**

1. Strong empirical performance and generalization. MMSeg achieves consistent improvements over both training-free and training-based baselines on diverse datasets (PerSeg, FSS, COCO-20i, PACO-Part), showing strong robustness and cross-domain generalization.

2. Well-structured design with effective ablation validation. The modular components (VLDM, TLGP, and COMP) are clearly defined, empirically justified through comprehensive ablation studies, and intuitively aligned with the paper’s motivation of enhancing semantic localization and mask refinement.

3. The paper is clear and easy to follow. The presentation is well-organized, with intuitive figures and a logical flow from motivation to experiments, making the overall work accessible and easy to understand.

**Weaknesses:**

1. Heavy reliance on multiple large pretrained models. Although the framework is training-free, MMSeg depends on several heavyweight vision foundation models (DINOv2, CleanDIFT, CLIP-Surgery, RAM, and SAM). This multi-model pipeline increases inference complexity and memory cost, which may limit its applicability in real-time or resource-constrained scenarios.

2. Lack of unified optimization and adaptive coordination. The three modules (VLDM, TLGP, and COMP) operate independently with heuristic fusion and fixed thresholds, without a joint optimization objective. This modular design may hinder global consistency and makes the method sensitive to hyperparameter tuning.

3. Incomplete or insufficient citation coverage. While the paper discusses related training-free segmentation and SAM-based prompting works, it overlooks several recent advances in multimodal or diffusion-based segmentation [1-4]. The lack of citation and comparison with some contemporaneous studies may weaken the positioning of MMSeg relative to concurrent approaches.

[1] Zhang R. et al. Personalize Segment Anything Model with One Shot. ICLR 2024.
[2] Hu J., Gong S., & Zhang Q. ProMaC: Prompt-aware Mask Calibration for Training-free Segmentation with Vision Foundation Models. NeurIPS 2024.
[3] Shang G. et al. Prompt-Driven Referring Image Segmentation with Instance Contrasting. CVPR 2024.
[4] Yang J. et al. Diffusion-Enhanced Cross-Modal Semantic Segmentation. ACM MM 2025.

**Questions:**

1. Parameter Sensitivity. The method introduces several hyperparameters (e.g., thresholds). How sensitive is MMSeg to these values?
Have the authors evaluated whether the segmentation quality degrades significantly when these are changed or tuned across datasets?

2. Pseudo-label Robustness. Since TLGP relies on pseudo-classes generated by RAM, what happens when the pseudo-labels are incorrect or ambiguous? Do the authors observe cascading errors in text-driven localization, and how does MMSeg mitigate such cases?

3. Diffusion Prior Effectiveness. The paper employs CleanDIFT to extract diffusion priors. Can the authors provide quantitative evidence that these low-level diffusion features improve performance beyond what DINOv2 or SAM features already offer? For example, what is the mIoU gain if CleanDIFT is replaced with a simpler visual encoder?

4. Comparison Scope and Missing References. The related work section omits several recent multimodal or diffusion-based segmentation studies [1–5]. Could the authors comment on how MMSeg compares conceptually and empirically with these methods, especially ProMaC [2] and Prompt-RIS [3] that also exploit multimodal consistency?

---

### Official Review · Reviewer_MazH · 2025-11-01

**Soundness:** 3
**Presentation:** 3
**Contribution:** 3
**Rating:** 6
**Confidence:** 4

**Summary:**

Visual prompt segmentation often suffers from granularity misalignment and imprecise localization due to its reliance on high-level features extracted by models such as DINOv2 and SAM. While text prompt segmentation can mitigate granularity inconsistencies, it struggles to handle targets with subtle or complex attributes, limited by linguistic ambiguity or insufficient descriptive capacity.
To address these issues, the paper proposes a training-free Multi-modal and Multi-view Segmentation framework (MMSeg), which integrates pseudo-class generation into the image-prompt segmentation pipeline. For localization optimization, it enriches feature diversity through diffusion priors, multi-view image augmentation, and pseudo-label generation. For mask optimization, it first samples point prompts to generate candidate masks, then filters and merges these masks via a multi-step consensus-oriented process. Extensive experiments validate that this method achieves excellent performance across multiple datasets and tasks. It supports one-shot semantic segmentation and generalizes well to one-shot part segmentation. Comprehensive ablation studies further confirm the effectiveness of each component in the framework.

**Strengths:**

1、This paper is well-structured, with the research motivation clearly elaborated in the introduction. It systematically summarizes the advantages and disadvantages of vision-prompt-based segmentation and text-prompt-based segmentation, and innovatively integrates these two approaches—making it a highly promising research idea.

2、The paper proposes a novel method for proposal points generation. By fusing visual information and pseudo-label category information, it better identifies point prompts for segmentation. Additionally, the methodology section is described in a clear and comprehensible manner.

3、In the COMP (Consensus-Oriented Mask Proposer) module, the paper presents an approach that fuses multiple point prompts and generates a more refined segmentation map.

4、The method proposed in the paper not only enables general segmentation but also achieves excellent performance in hierarchical segmentation tasks (e.g., Pascal Part). This versatility is conducive to the practical promotion and application of the method.

**Weaknesses:**

1、The method seems to utilize a variety of foundation models, such as diffusion models, RAM, SAM, and CLIP. Therefore, how does the efficiency of the entire framework compare to other methods? Will it be too slow? It is also necessary to conduct an ablation study on the time consumption of each module.

2、The method performs classification via RAM (or more precisely, CLIP-based approaches) and then identifies regions belonging to the target category in the image through comparison for localization. This aspect is highly dependent on the feature alignment capability of vision-language models (e.g., CLIP). Additionally, it requires a predefined category vocabulary, which prevents truly open-vocabulary classification—this is unfavorable for training-free scenarios. Existing generative classification methods, such as VLMs (Vision-Language Models), serve as a reference localization approach that identifies correlations between reference images and test images through multi-image input. Please discuss the advantages and disadvantages of the proposed method in this paper compared to this paradigm.

3、Are the methods compared in the paper outdated? For instance, the training-based methods only include SegGPT from 2023.

4、As we know, SAM achieves segmentation by treating points that hit the target as positive examples and points that do not hit the target as negative examples. It is necessary to conduct an ablation study on the accuracy of positive examples hitting the target and negative examples not falling on the target during the points generation process, so as to demonstrate the quality of localization.

**Questions:**

See weakness part.

---

### Note · Authors · 2025-11-12

I have read and agree with the venue's withdrawal policy on behalf of myself and my co-authors.